# What do the sustainable development goals reveal, and are they sufficient for sustainable development?

**Bing Wang, Tianchi Chen** *

College of Public Administration, Huazhong University of Science and Technology (HUST), Wuhan, China

* d202081314@hust.edu.cn

**Data Availability Statement:** All relevant data are within the manuscript and its Supporting Information files.

**Funding:** The paper has been supported by the Key Projects of the National Social Science Foundation

## Abstract

The Sustainable Development Goals (SDGs) adopted by the United Nations in 2015 represent the current perceptions of humans regarding understanding and monitoring development. Achieving all 17 goals simultaneously is unrealistic. Considering the interconnected nature of SDGs, identifying their critical dimensions, goals, indicators, and mutual relationships is necessary. In addition, with increasing reservations about the sustainability of SDGs, it is crucial to explore consistency across different dimensions to ensure policy coherence in maximizing synergies and minimizing trade-offs. Our study employed multiple factor analysis (MFA) and hierarchical clustering on principal components (HCPC) to investigate these issues and analyze the results based on the public value (PV) theory. The results indicated that the Human Development Index (HDI) and gross domestic product per capita (GDPP) constitute the first principal component (PC) and are determinants in differentiating country clusters. However, they contradict environmental indicators such as $CO_2$ emissions per capita and ecological footprint gha per person (EFP) and have low synergy with the Happy Planet Index (HPI). Additionally, the relationships between income level, inequality, and environmental quality correspond to a combined Kuznets curve and an environmental Kuznets curve (EKC). Moreover, governance capacity has become increasingly crucial in sustainable development, particularly in the capability to prioritize different PVs in a timely and strategic manner. Finally, despite the novelty of EFP and HPI, they cannot reveal the entire development story. SDGs require embracing more such indicators to enrich the value bases of development and achieve a sustainable future.

## 1. Introduction

With the rapid advancement of economies and technology, a continuously changing understanding, theories, and practices of human development have been witnessed over the past several decades. In the 1940s, with the invention of the gross domestic product (GDP), economic growth was equated with modernization, guided by theories of growth and modernization [1]. Afterward, since the 1960s, as the dependency theory defined development as social progress, equitable growth and basic human needs such as poverty, education, health, and social welfare

(23AZD013). The funders had no role in study design, data collection and analysis, decision to publish, or preparation of the manuscript.

**Competing interests:** The authors have declared that no competing interests exist.

**Abbreviations:** SDGs, Sustainable Development Goals; GDP, gross domestic product; GDPP, gross domestic product per capita; HDI, Human Development Index; EF, ecological footprint; EFP, ecological footprint gha per person; HPI, Happy Planet Index; PV, public value; MFA, multiple factor analysis; HCPC, hierarchical clustering on principal components; PCA, principal component analysis; PC, principal component; EKC, the environmental Kuznets curve.

were integrated into the development model [2, 3]. For instance, Sen's entitlement theory defines development as expanding real freedoms inspired the invention of the Human Development Index (HDI) in 1990 [4, 5].

Since the 1970s, the constant conquering of nature to achieve economic growth and social welfare has led to increasingly intense conflicts between humans and nature, resulting in global warming, excessive deforestation, environmental pollution, urban expansion, and energy exhaustion [6]. To address these development dilemmas, "sustainable development" was first proposed in the Brundtland Report in 1987, defining sustainable development as "meeting the needs of the present without compromising the ability of future generations to meet their own needs" [7]. "Sustainable development" expanded the dimension of development to a synchronous development of the economy, society, and the environment [8, 9]. In September 2015, 193 UN member states unanimously adopted the 2030 Agenda for Sustainable Development at the UN General Assembly Summit. The agenda is a plan of action for people, planet, prosperity, peace, and partnership based on a holistic conception of development and provides a set of Sustainable Development Goals (SDGs) committed to establishing a global strategic roadmap to end poverty, provide a dignified living, and secure the planet under the pledge that no one will be left behind [10].

The SDGs contain 17 goals, 169 targets, and 232 indicators (more than 650 if subdivisions are included) covering the economic, social, environmental, and governance dimensions. In adopting the 2030 Agenda, the UN stressed that the interconnected and integrated nature of the SDGs is essential for ensuring the attainment of objectives [10]. Many studies have investigated the synergy and trade-offs among goals, targets, and indicators. However, no consensus has yet been reached [11]. Although the SDGs constitute the most comprehensive assessment framework for sustainable development, an increasing number of studies have indicated the inherent contradictions between socioeconomic and environmental goals, as well as the Earth's carrying capacity [12–14]. Consequently, this study proposed the following two research questions:

Q1: What are the main dimensions, objectives, and indicators of the SDGs, and how do they relate to each other?

Q2: Are SDGs sufficient to support a sustainable future?

To answer these questions, we employed multiple factor analysis (MFA) and hierarchical clustering on principal components (HCPC) to conduct a global-scale statistical analysis of the SDGs and analyzed the results based on public value (PV) theory. Additionally, to explore the relationship between SDGs, income level, and their alignment with environmental goals and the Earth's carrying capacity, four illustrative groups (IG) were chosen for comparison but were not involved in the MFA and HCPC. Specifically, they were IG 1 (GDP per capita, GDPP), IG 2 (Human Development Index, HDI), IG 3 (ecological footprint per person, EFP), and IG 4 (Happy Planet Index, HPI).

Our analysis provided four results. (1) HDI and GDPP relate closely to SDGs and are determinants in differentiating country clusters. Conflicts exist between socioeconomic and environmental goals and Earth's carrying capacity. Economic growth is achieved at the cost of increased per capita carbon emissions and per capita ecological footprint (EFP), with no evident gain in HPI. (2) Countries' projections on the first plane of the MFA approximately correspond to a combined Kuznets curve and an environmental Kuznets curve (EKC). (3) As an increasingly important dimension and means of achieving sustainable development, governance capacity is a critical enabler of SDGs. Governments should strengthen cooperation across sectors and form holistic strategies to prioritize public values (PV) to avoid development

failures. (4) Neither EFP nor HPI are comprehensive, considering that human development is based on multidimensional PVs. What matters is to "jump out of the box," exploring and embracing new theories, values, and indicators, which enables us to avoid development failures and achieve a sustainable future.

The remainder of this paper is organized as follows. The second section is a systematic review of the value and PV, the PV foundation of GDP and "beyond GDP" indicators, and interlinkages of the SDGs. The third section explains the data preprocessing and adequacy tests in detail. The fourth section interprets the methodology employed in this study, including the MFA and HCPC. The fifth section discusses the results. Finally, the sixth section concludes the paper and proposes several policy suggestions and research limitations.

## 2. Literature review

### 2.1 Value and public value

Value is the formal academic concept of goodness. Kluckhohn [15] defined value as "a conception, explicit or implicit, distinctive of an individual or characteristic of a group, of the desirable which influences the selection from available modes, means, and ends of action." Values exist in all social spheres and profoundly impact our perception of reality, such as what is good or bad; provide identity for individuals, groups, and organizations; and guide our behavior [16]. Public value (PV) inherits from the concept of value, referring to values shared by the public, such as wealth, fairness, freedom, democracy, love, happiness, and technology [17]. Governments have created and safeguarded various PVs through services, laws, and regulations. Thus, PV forms the basis for guiding government governance and formulating public policies.

Neoliberal economics, which has dominated Western development theory, considers development based on value monism and the human nature assumption of homo economicus, whereas real human society is based on value pluralism, including homo sociologicus and homo politicus [18–20]. Owing to the incomparability and incommensurability of pluralistic values, defining and evaluating a wide range of developmental phenomena using monistic values is misleading. Considering the value of pluralism, several studies have classified PV into different forms. For example, Meynhardt [21] related basic needs to value dimensions and classified PVs as moral-ethical, hedonistic-esthetic, political-social, and utilitarian-instrumental. Benington [20] extended PVs beyond markets and the economy, identifying that the economy, politics, society, culture, and ecology all have PVs. Papi et al. [22] divided PVs into social, economic, and intangible values.

With the development of society, an increasing number of value dimensions that share intimate and complex relationships are emerging. One typical feature is the hierarchy of PVs, in which some values are more basic and precede others [23], such as economic growth in the early stage of national development. Another form is the conflict between the PVs. Some PVs, such as growth and happiness, are consistent, but sometimes, realizing some values may cause conflicts with others [24, 25]. For instance, economic growth may lead to environmental deterioration and widen the gap between the rich and poor. Economic growth at the expense of other PVs, such as equity, health, and the environment, is commonplace but cannot be considered sustainable development. In this era of unprecedented economic affluence accompanied by social and ecological crises, PV governance has been regarded as a new theoretical paradigm that goes beyond traditional and new public management. It can address the problems of pluralistic social development and help achieve sustainable development [26–28].

## 2.2 Public value foundation of GDP and "beyond GDP" indicators

GDP was invented in the 1930s during the Great Depression and World War II. Since the 1940s, it has become one of the most influential development indicators promoted by the theories of growth and modernization. Modernization refers to the gradual transition from a "traditional" to a "modern" society, which could be achieved through the adoption of Western cultural and institutional practices [29]. Therefore, the core objective of developing countries is to catch up with developed industrialized countries through rapid GDP growth. In this context, the GDP has been regarded as one of the most prominent indicators for measuring a country's development and was honored as the greatest invention of the 20th century [30, 31]. The GDP and its represented economic values have been regarded as the most critical PVs, stimulating post-war reconstruction and economic recovery. At that time, promoting economic growth was almost the primary policy goal of all governments. As measured by GDP, economic growth was generally considered equivalent to social progress and modernization in the 1950s and the 1960s [32].

However, the use of GDP as a proxy for human development has attracted widespread criticism. Not all social phenomena and human behaviors can be monetarized as a reference to market prices, such as social justice, individual mental health, and natural resources that money cannot fully measure. The economic way of thinking has earned a reputation for knowing the price of everything but the value of nothing [33]. As many valuable items are excluded and materialized by GDP, we appear to be richer than we really are, while the reality is that we are losing many valuable values and resources [34].

Since the 1960s, the concept of development has broadened to many other PV dimensions, particularly social values related to basic human needs, such as poverty, education, health, and social welfare, which have introduced some prominent indicators, such as the Physical Quality of Life Index (PQLI) [35] and the well-known Human Development Index (HDI) [4, 5, 36]. The HDI is the best-known alternative to GDP for measuring human development and is composed of income, health, and education dimensions, although it has often been criticized for its incomplete assessment of human development and high correlation with GDP [37–39].

Since the 1970s, under the intense conflicts between human and ecological systems, the Brundtland Report [7] systematically discussed the connotation of "sustainable development." Social development has expanded to dimensions of economy, society, and the environment, resulting in the emergence of indicators focused on environmental values and those comprising multiple dimensions, such as the Ecological Footprint (EF) [40], Millennium Development Goals (MDGs), Sustainable Development Goals (SDGs), and Happy Planet Index (HPI) [41]. The EF measures the need of humans for biocapacity, providing an integrated, multiscale approach for measuring human consumption and the overshoot of natural resources [42], and has been widely employed to measure the pressure caused by human activities on Earth [43–45]. The HPI measures sustainable well-being worldwide, ranking and comparing how efficiently residents of different countries use environmental resources to lead long and happy lives [46]. Significantly, it creatively proposes the concept of a "well-being economy" that expands the EF and reveals the balance between human well-being and the Earth's carrying capacity.

Considering the extensive use of GDPP, HDI, EF, and the originality of HPI, they were selected as illustrative indicators to explore the interlinkages of the SDGs in our study.

## 2.3 Research on the interlinkages of the SDGs

The interlinkages of SDGs among goals, targets, and indicators have recently become a research hotspot. There are three interlinkage types: synergies, trade-offs, and neutral

relationships [47]. The achievement of the 2030 Agenda depends primarily on whether countries can maximize synergies while reducing the negative impacts of trade-offs. Most studies have adopted quantitative approaches to identify interlinkages on the scales of global, regional, and intergovernmental organizations. The three primary research techniques are summarized as follows.

First is semi-quantitative cross-impact analysis [47–50]. For example, McCollum et al. [48] conducted a large-scale cross-impact assessment of energy-related interactions between SDGs, and Dawes [50] employed a dynamic model to investigate the direct and indirect effects of each goal and their impacts on others. Second, network analysis is the most popular approach in this field [49, 51–53]. For example, Bali Swain and Ranganathan [52] modeled the interlinkages between 17 goals in OECD, South Asia, Latin America, Sub-Saharan Africa, and Middle Eastern and Northern Africa (MENA) countries, while Laumann et al. [53] considered goals as network nodes and attempted to find differences between country groups. Third, statistical analysis techniques are widely employed in investigating SDGs interlinkages, including correlation analysis [54–56], structural equation models [57] and dimension reduction analysis [58, 59]. For example, ICSU [54] examined 316 interactions regarding SDGs 2, 3, 7, and 14 at goal and target levels via correlation analysis; Spaiser et al. [58] adopted confirmatory factor analysis (CFA) and principal component analysis (PCA) to explore synergies and trade-offs between economic growth, socioeconomic, and environmental goals; and Cling & Delecourt [59] adopted multiple factor analysis (MFA) to explore the interlinkages of SDGs at goal and indicator levels.

In summary, most studies have demonstrated that synergies are more substantial than trade-offs. However, socioeconomic goals are generally mutually reinforced; in contrast, they usually contradict environmental goals. As Wackernagel et al. [12] suggested, "SDGs expressed today vastly underperform on sustainability, especially regarding equity challenges and physical constraints imposed by planetary limits." Therefore, this study aimed to identify key dimensions and investigate the interlinkages of the SDGs as well as to explore whether the SDGs are sufficient for a sustainable future.

## 3. Data

### 3.1 Data source, selection criteria, and missing data imputation

In our study, we obtained data from the official SDG Global Database [60] and the World Development Indicators (WDI) database [61]. Although both databases provide data from 1990 to 2021, we could not conduct a time-series analysis due to the large amount of missing indicator data at different time points. Moreover, due to the impact of the global COVID-19 pandemic, missing data after 2019 was excessive, and the data fluctuated significantly compared with 2019 (such as 8.1.1: annual growth rate of real GDP per capita and 9.1.2: air transport, passengers carried). Therefore, a cross-sectional analysis was conducted based on the 2019 data. We built the largest possible database by respecting the criteria and rigorous screening steps required for our statistical study (S1 Appendix).

Finally, we retained 133 countries (S3 Table) and 78 indicators (excluding four illustrative indicators: GDPP, HDI, EFP, and HPI) (S1 Table) for MFA and HCPC. Among the 78 indicators, 68 (87%) were from the UNSDG database, and 10 (13%) were from the World Bank WDI database. Meanwhile, 76 (97%) were Tier 1 indicators, 2 (3%) were Tier 2 indicators, and no Tier 3 indicators were used. According to the tier classification defined by IAEG-SDGs updated on 09 June 2022, tier 1 indicators are those with clear concept, internationally established methodology and standards, and regularly produced by at least 50% countries and population in every region. Compared with tier 1, the only difference of tier 2 indicators is that

data are not regularly produced by countries. Tier 3 indicators are those with no available international established methodology or standards.

## 3.2 Data normalization and adequacy tests

Before proceeding with the MFA, an essential step was to transform the selected 78 indicators and the four illustrative ones into dimensionless and comparable numbers. In general, several primary normalization methods exist, such as Min-Max, Z-score, and Decimal Scaling normalization. Our study used Min-Max normalization, as it provides a linear transformation of the original data between 0 and 1. In addition, it standardizes 37 negative indicators (including illustrative ones; see S1 Table) to ensure that higher values indicate better performance.

After data normalization, the final step was to test whether the normalized data were suitable for MFA using the Kaiser–Meyer–Olkin (KMO) and Bartlett's tests of sphericity. Kaiser [62] recommended a minimum KMO value of 0.5, and the current consensus is that a value between 0.5 and 0.7 is considered mediocre, a value between 0.7 and 0.8 is good, and a value between 0.8 and 0.9 is considered great. If the KMO value exceeds 0.9, the data are considered excellent for factor analysis [63]. Additionally, Bartlett's test of sphericity suggests whether the chosen variables are sufficiently correlated to conduct factor analysis. Bartlett [64] asserted that the significance should be less than the value of 0.05 ($p < 0.05$) for the factor analysis to be appropriate. For our chosen data, the KMO value was 0.84, as shown in Table 1, and the p-value calculated from Bartlett's test of sphericity was extremely small (approximately zero), indicating high correlations. Therefore, the chosen data passed the adequacy tests, and we proceeded with MFA.

## 4. Methodology

In our study, Multiple Factor Analysis (MFA) and Hierarchical Clustering on Principal Components (HCPC) were employed as primary tools in investigating the two research questions mentioned before. That is, what do the SDGs reveal, and are they sufficient to support a sustainable future?

## 4.1 Multiple Factor Analysis (MFA)

MFA is designed to handle multiple data tables that measure sets of variables collected on the same observations [65]. As an extension of the principal component analysis (PCA), it is a popular tool for dealing with double-counted and overlapping information in two or more variables [66]. MFA was performed in two steps. First, PCA was conducted for each set of SDG, which was then normalized by dividing all its elements by the square root of the first eigenvalue obtained from its PCA. According to Abdi et al. [67], this step is crucial to ensure that the length of the first principal component (PC) of each SDG is equal to one; thus, no SDG can dominate the common solution because it has larger inertia in its first dimension. Second, all normalized SDGs were concatenated into a grand data table, and then a non-

**Table 1. KMO and Bartlett's tests of sphericity.**

| KMO test (Overall MSA) | | **0.84** |
|---|---|---|
| Bartlett's test of sphericity | chisq | 12244.61 |
| | p-value | 0.000 |
| | df | 3321 |

Source: author's calculations.

normalized PCA (global PCA) was performed to obtain a set of factor scores for countries and loadings for indicators. Subsequently, the SDGs were projected onto the global analysis to identify communalities and discrepancies, allowing the exploration of correlations among goals, indicators, and countries (for a systematic explanation of MFA's principles, see [67–69]. Notably, SDG 3 had 18 indicators, occupying almost a quarter of the selected indicators, whereas SDG 4, 5, 6, and 11 comprised only three indicators. MFA considered the unequal size of groups and ensured that no SDGs could outweigh the influence of others in determining the first axis of the PCA.

To initiate the MFA, we employed the "FactoMineR" [70] and "factoextra" [71] packages using software R [72]. We started the MFA by identifying and selecting the PCs. The PC refers to a new variable constructed by a linear combination of the original indicators, and its importance is reflected by the eigenvalue, which suggests that the total variance is explained. The first PC maximizes the variance, explaining the most significant amount of information in the SDGs, and the second PC accounts for the second-highest variance. All the PCs were orthogonal, suggesting mutual independence. Because 78 indicators were used as variables in the analysis, the MFA generated 78 PCs, but only those with high eigenvalues (usually higher than 1) were selected as PCs. In our study, the first eight PCs satisfied the criterion, and half exceeded 1.5. In addition, using the scree test method [73] demonstrated in Fig 1, the first four PCs explained a total variance of 45.7% and were selected as the target PCs.

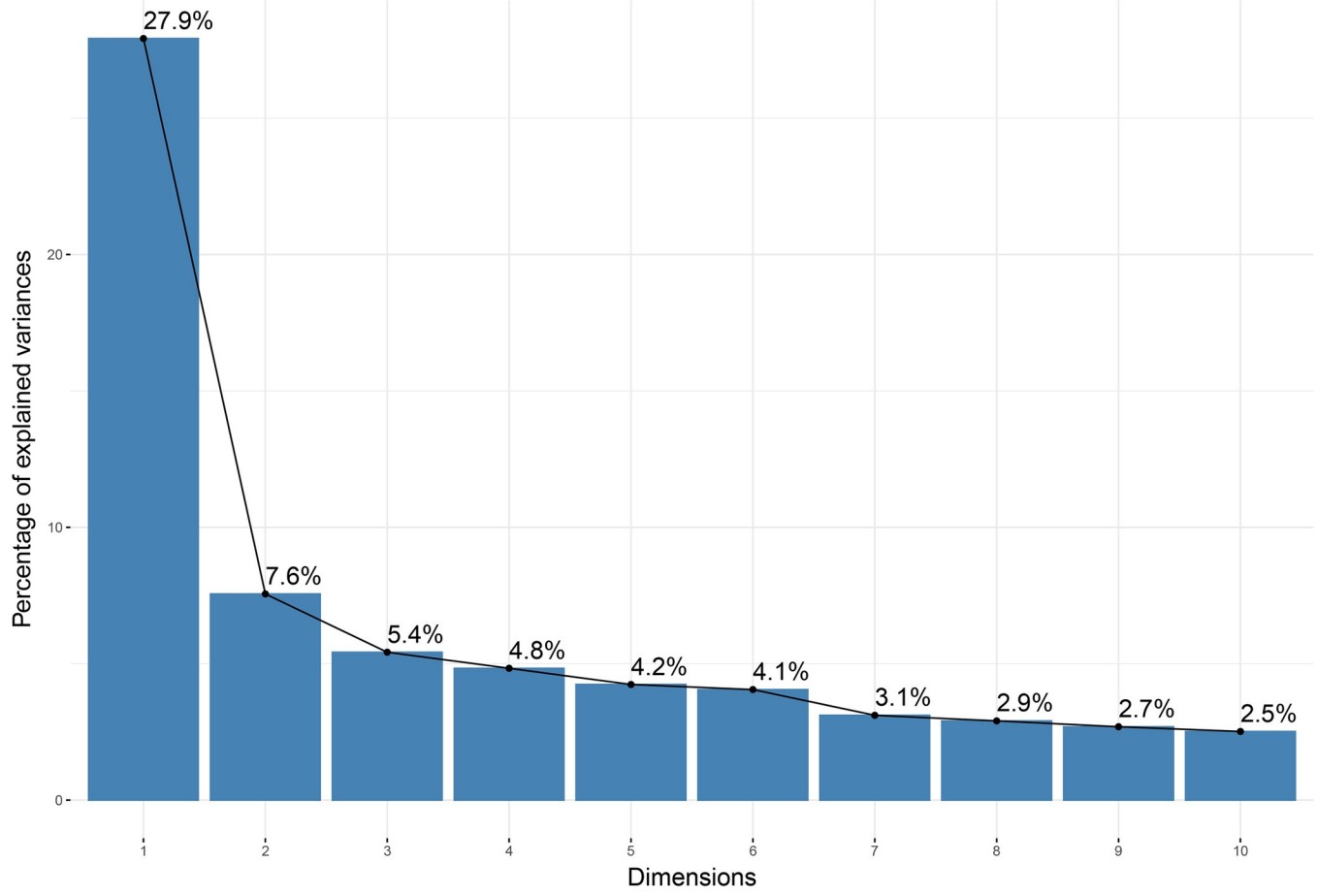

**Fig 1. Scree plot.**

**Table 2. Contributions of SDGs to the selected four PCs.**

| PC 1 | | PC 2 | | PC 3 | | PC 4 | |
|---|---|---|---|---|---|---|---|
| SDGs | Contrib. (%) | SDGs | Contrib. (%) | SDGs | Contrib. (%) | SDGs | Contrib. (%) |
| SDG 3 | 9.18 | SDG 10 | 15.49 | SDG 16 | 21.20 | SDG 8 | 40.79 |
| SDG 2 | 8.96 | SDG 15 | 12.62 | SDG 15 | 13.57 | SDG 10 | 7.98 |
| SDG 17 | 8.67 | SDG 6 | 11.78 | SDG 5 | 13.19 | SDG 15 | 7.58 |
| SDG 1 | 8.52 | SDG 5 | 11.69 | SDG 6 | 11.35 | SDG 5 | 7.55 |
| SDG 4 | 8.16 | SDG 12 | 9.96 | SDG 10 | 10.76 | SDG 16 | 7.27 |
| SDG 7 | 7.17 | SDG 11 | 9.37 | SDG 4 | 4.60 | SDG 6 | 6.01 |
| SDG 11 | 6.81 | SDG 9 | 6.44 | SDG 8 | 4.44 | SDG 17 | 4.49 |
| SDG 16 | 5.85 | SDG 7 | 5.05 | SDG 17 | 3.94 | SDG 9 | 3.99 |
| SDG 10 | 5.77 | SDG 16 | 4.51 | SDG 7 | 3.85 | SDG 12 | 3.51 |
| SDG 5 | 5.72 | SDG 1 | 2.92 | SDG 1 | 3.78 | SDG 7 | 2.73 |
| SDG 9 | 5.71 | SDG 17 | 2.58 | SDG 9 | 3.68 | SDG 13 | 2.58 |
| SDG 13 | 5.60 | SDG 3 | 2.56 | SDG 3 | 2.00 | SDG 2 | 2.42 |
| SDG 12 | 5.44 | SDG 8 | 1.50 | SDG 12 | 1.38 | SDG 1 | 0.93 |
| SDG 8 | 4.36 | SDG 13 | 1.27 | SDG 11 | 1.21 | SDG 3 | 0.79 |
| SDG 6 | 2.79 | SDG 4 | 1.20 | SDG 2 | 0.93 | SDG 11 | 0.73 |
| SDG 15 | 1.31 | SDG 2 | 1.05 | SDG 13 | 0.13 | SDG 4 | 0.66 |
| Explained variances (%) | 27.9 | | 7.6 | | 5.4 | | 4.8 |

Then, we adopted "contributions" to define PCs and explore relationships between PCs, indicators, and countries. The concept of "contributions" relates to "factor scores" and "loadings." The former denotes the countries' coordinates, which are the projections of the countries on the PCs, whereas the latter refers to the indicators' coordinates on the PCs computed by the correlations between the PCs and indicators. According to Abdi et al. [67], a country's contribution to a PC is measured as the ratio of the squared weighted factor score to the dimension eigenvalue, and the contribution of an indicator to a PC is evaluated as its squared weighted loading for this PC. Finally, because each goal comprises many indicators, its contribution to a PC can be calculated as the sum of its indicator contributions. The contribution of a country, indicator, goal, or PC ranges from 0% to 100%, and the sum of the contributions of all its elements is 100%. Table 2 summarizes the contributions of each goal to the first four PCs. Additionally, we adopted RV coefficients to measure the similarity among the 17 SDGs and four illustrative groups (Fig 2), as this is a widely used tool to assess the similarity among datasets by reflecting the variances shared by two or more matrices [74].

## 4.2 Hierarchical Clustering on Principal Components (HCPC)

HCPC is an effective data mining method for discovering knowledge in multivariate datasets that identifies clusters of similar objects within a dataset (or multiple datasets) of interest. The HCPC combines three standard methods employed in multivariate data analysis: principal component methods, hierarchical clustering, and partitioning clustering, particularly the k-means method [75]. Hierarchical clustering was performed using Ward's criterion [76] for the first four PCs. The following procedure was used to determine the cluster partitioning:

1. The sum of the within-cluster inertia was first calculated for each partition;

2. The optimal division was the one with the higher relative loss of inertia using the formula $\frac{i(cluster(n+1))}{i(cluster(n))}$, where n refers to the number of suggested partitions;

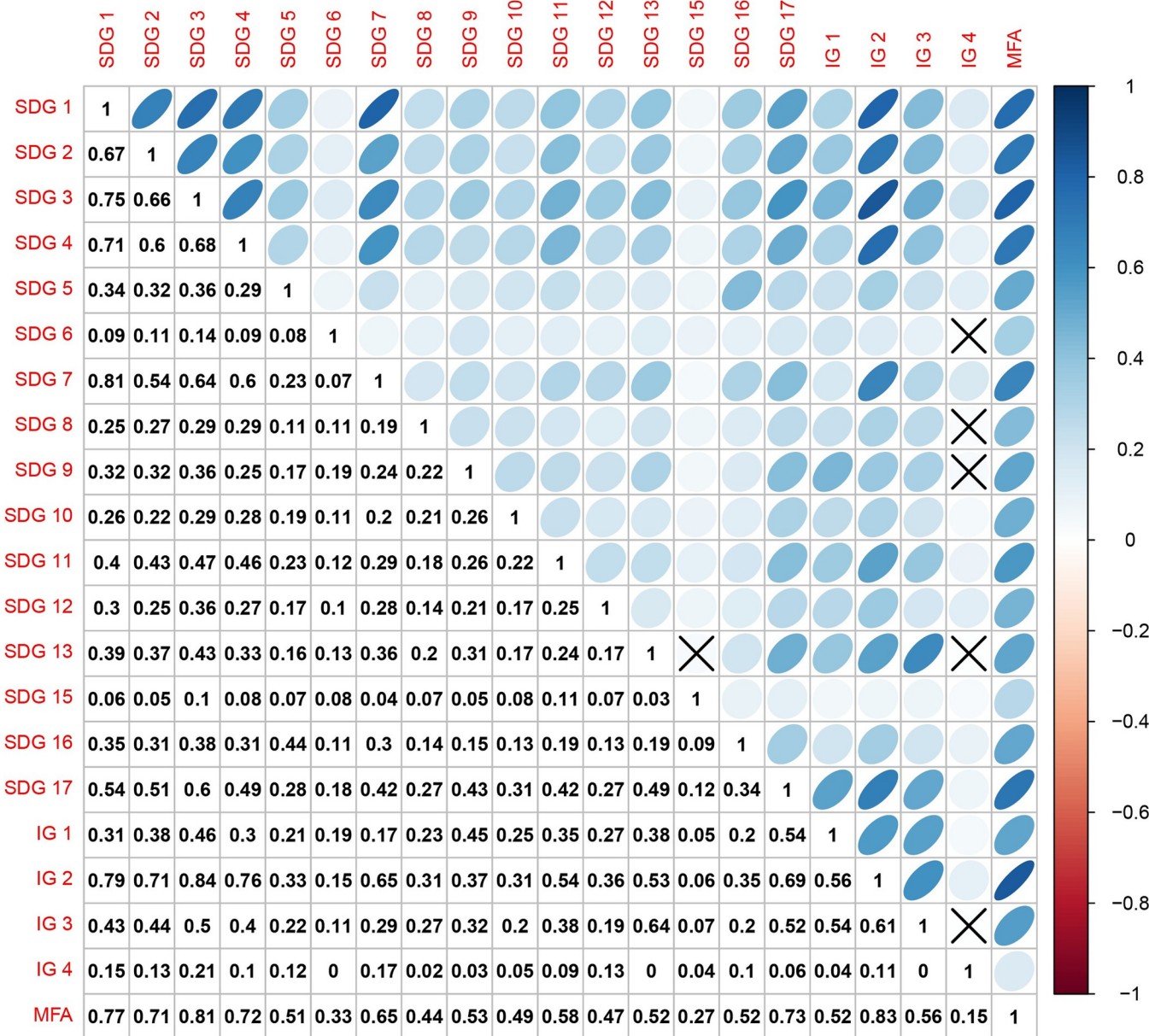

**Fig 2. RV coefficients.** Note: RV correlations that did not pass the significance test (0.05) are marked as X.

3. The partitioning was made more robust by applying k-means consolidation. Specifically, we conducted a maximum of 50 iterations for consolidation, making it possible to find the clustering with the highest between-group variance.

Finally, the algorithm generated three clusters containing 38, 58, and 37 countries, and their GDPP and HDI averages were calculated (S3 Table).

## 5. Results and discussion

As an up-to-date dynamic monitoring framework that measures sustainable development across countries, SDGs comprised a more comprehensive public value (PV) base, considering

that the first PC explained only 27.9% of the variance, whereas the other PCs explained the remaining variances in similar proportions. Specifically, the latter five PCs explained 7.6%, 5.4%, 4.8%, 4.2%, and 4.1% of the variance, respectively. This section sequentially presents our findings based on the first four PCs.

## 5.1 HDI and GDPP relate closely to SDGs

The Scree plot (Fig 1) demonstrated that the first PC explained 27.9% of the total variance, while SDGs 1–4 and 17 were top contributors (Table 2). SDGs 1–4 represent no poverty, zero hunger, good health and well-being, and quality education, respectively. In comparison, HDI comprises three dimensions of human development: a long and healthy life, knowledge, and decent living. Therefore, the primary contributors to the first PC had information similar to that of HDI. In addition, as Fig 2 illustrates, the RV correlations between HDI (IG 2) and SDGs 1–4 and 17 were moderately high at 0.79, 0.71, 0.84, 0.76, and 0.69, respectively. Furthermore, the correlation value between HDI (IG 2) and the first PC was 0.97 (S2 Table), indicating that HDI could explain 94% ($R^2$) of the first PC.

A more detailed correlation analysis was conducted to investigate the relationships between GDPP, HDI, and the first PC (S2 Table and S1 Fig). GDPP (IG 1) had a correlation value of 0.74 with the first PC, and the correlation value between GDPP and HDI reached 0.96, suggesting that GDPP could explain 92% ($R^2$) of the HDI, proving their similarity. Therefore, the GDPP also played a critical role in the first PC, explaining why SDG 17 contributed the third most to the first PC. Specifically, SDG 17 comprised indicators that closely correlated with GDP, such as total government revenue as a proportion of GDP (%) (17.1.1) and foreign direct investment (FDI) inflows (17.3.1). Similar to SDGs 1–4 and 17, SDG 7 (affordable and clean energy) and SDG 11 (sustainable cities and communities), which comprise indicators relevant to basic human needs, also contributed considerably to the first PC and had intimate relationships with HDI and GDPP (Fig 2). Therefore, considering the strong correlations between the first PC, HDI, and GDPP, the first PC could be defined as an economic development dimension.

As shown in Table 3, indicators that contributed most to the first PC mainly included 13.2.2 ($CO_2$ emissions-metric tons per capita), 11.1.1 (population living in slums), 4.1.2 (completion rate, lower secondary education), 17.8.1 (internet users per 100 inhabitants), 2.2.1 (prevalence of stunting, height for age), 17.6.1 (fixed internet broadband subscriptions per 100 inhabitants), 7.1.2 (proportion of the population with primary reliance on clean fuels and technology), 7.1.1 (proportion of pollution with access to electricity), 1.3.1 (proportion of the population covered by at least one social protection benefit). Notably, there were fewer indicators from SDGs 1 and 3, and the reason may lie in the principle of MFA, which automatically reduces the weights of SDGs that contain many highly correlated indicators.

The correlation circles in Fig 3 illustrate the 30 indicators with the highest contributions and their mutual relationships along the coordination axes constituted by the first four PCs. Highly correlated indicators converge closely with smaller angles, whereas weakly or negatively correlated indicators are positioned nearly perpendicular or opposite. Additionally, an indicator closer to the circumference of a circle indicates that it occupies a higher proportion or is more represented in the plane [75]. As demonstrated in the first circle constituted by the first two PCs, most indicators with high contributions densely surrounded the positive axis of the first PC, implying high similarities. This confirms the argument that SDGs are more interconnected than MDGs and that their synergistic effect is more significant than the trade-offs [52, 77]. HDI and GDPP were well represented by the first PC and formed sharp angles with most other indicators, demonstrating that most indicators synergize with a country's economic

**Table 3. Twenty indicators with the highest contributions to the first four PCs.**

| PC 1 | | PC 2 | | PC 3 | | PC 4 | |
|---|---|---|---|---|---|---|---|
| Indicator | Contrib. (%) | Indicator | Contrib. (%) | Indicator | Contrib. (%) | Indicator | Contrib. (%) |
| 13.2.2 | 5.60 | 6.4.2 | 10.76 | 16.1.1 | 11.45 | 8.1.1 | 16.97 |
| 5.2.1 | 5.14 | 5.5.2 | 8.79 | 6.5.1 | 10.85 | 8.2.1 | 16.92 |
| 11.1.1 | 4.57 | 12.c.1 | 7.87 | 10.4.2 | 7.50 | 5.5.1 | 7.06 |
| 16.9.1 | 3.64 | 11.6.2 | 6.95 | 5.5.1 | 7.09 | 8.5.2 | 6.77 |
| 4.1.2 | 3.43 | 15.1.1 | 6.59 | 16.7.1 | 6.99 | 15.5.1 | 5.73 |
| 17.8.1 | 3.27 | 10.4.1 | 6.39 | 5.5.2 | 5.12 | 16.7.1 | 4.64 |
| 2.2.1 | 3.26 | 15.1.2 | 5.73 | 15.5.1 | 5.03 | 10.4.2 | 4.19 |
| 8.10.2 | 2.97 | 9.4.1 | 5.68 | 15.1.2 | 4.15 | 6.4.2 | 4.06 |
| 17.6.1 | 2.96 | 10.a.1 | 4.19 | 4.5.1 | 4.03 | 12.c.1 | 2.94 |
| 7.1.2 | 2.85 | 10.4.2 | 3.51 | 15.1.1 | 3.84 | 9.2.1 | 2.70 |
| 2.2.3 | 2.77 | 7.2.1 | 2.81 | 10.7.4 | 3.17 | 13.2.2 | 2.58 |
| 12.2.2 | 2.72 | 5.5.1 | 2.63 | 9.5.1 | 2.48 | 17.1.1 | 2.18 |
| 6.5.1 | 2.67 | 11.5.1 | 2.37 | 16.9.1 | 2.15 | 16.1.1 | 2.08 |
| 10.4.1 | 2.65 | 17.1.2 | 2.34 | 7.1.1 | 1.90 | 10.7.4 | 2.07 |
| 2.1.1 | 2.60 | 16.7.1 | 2.28 | 8.5.2 | 1.70 | 17.1.2 | 1.74 |
| 4.2.2 | 2.60 | 12.b.1 | 1.82 | 8.1.1 | 1.56 | 7.3.1 | 1.62 |
| 7.1.1 | 2.43 | 16.1.1 | 1.54 | 17.10.1 | 1.52 | 10.4.1 | 1.41 |
| 11.6.2 | 2.21 | 7.1.1 | 1.53 | 1.1.1(b) | 1.17 | 2.1.1 | 1.40 |
| 4.5.1 | 2.13 | 10.7.4 | 1.40 | 11.6.2 | 1.17 | 6.6.1 | 1.27 |
| 1.3.1 | 1.97 | 13.2.2 | 1.27 | 7.2.1 | 1.11 | 15.1.1 | 1.22 |

development level. Significantly, 13.2.2 ($CO_2$ emissions per capita) contributed the most to the negative axis of the first PC, forming a sharp angle with EFP (its correlation value with the first PC was -0.78, S2 Table), suggesting that $CO_2$ emissions per capita and EFP share similar information. This result revealed that current global developments fail to constrain themselves

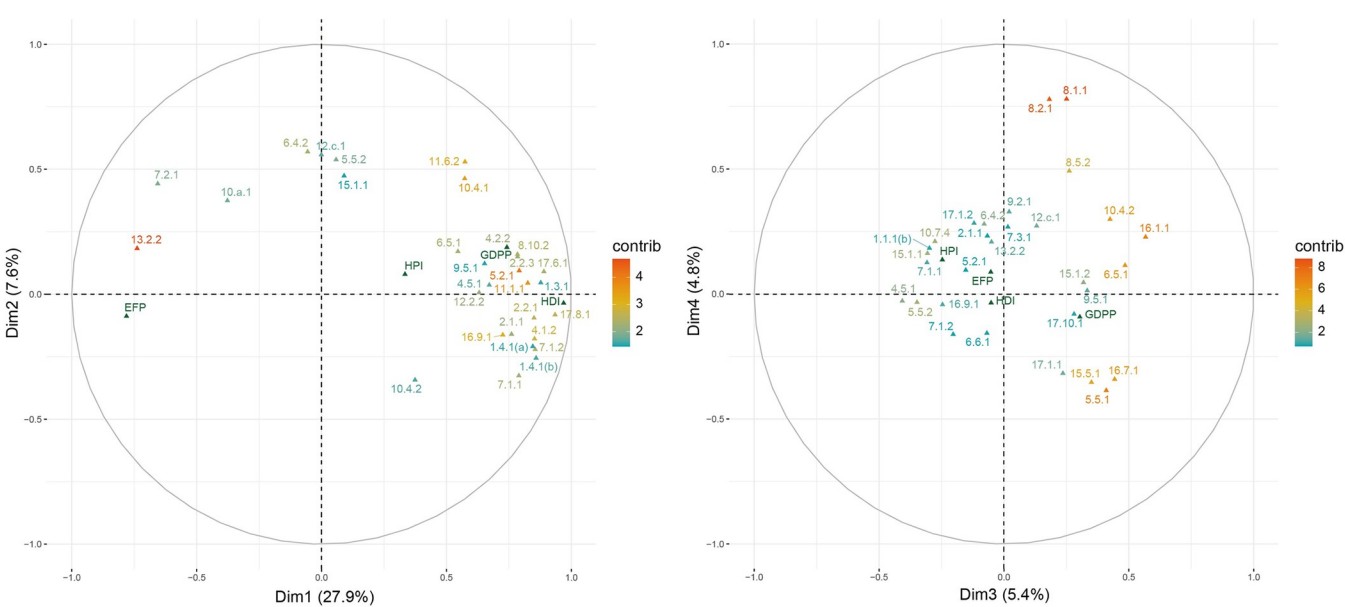

**Fig 3. Correlation circles with 30 indicators with the highest contributions.**

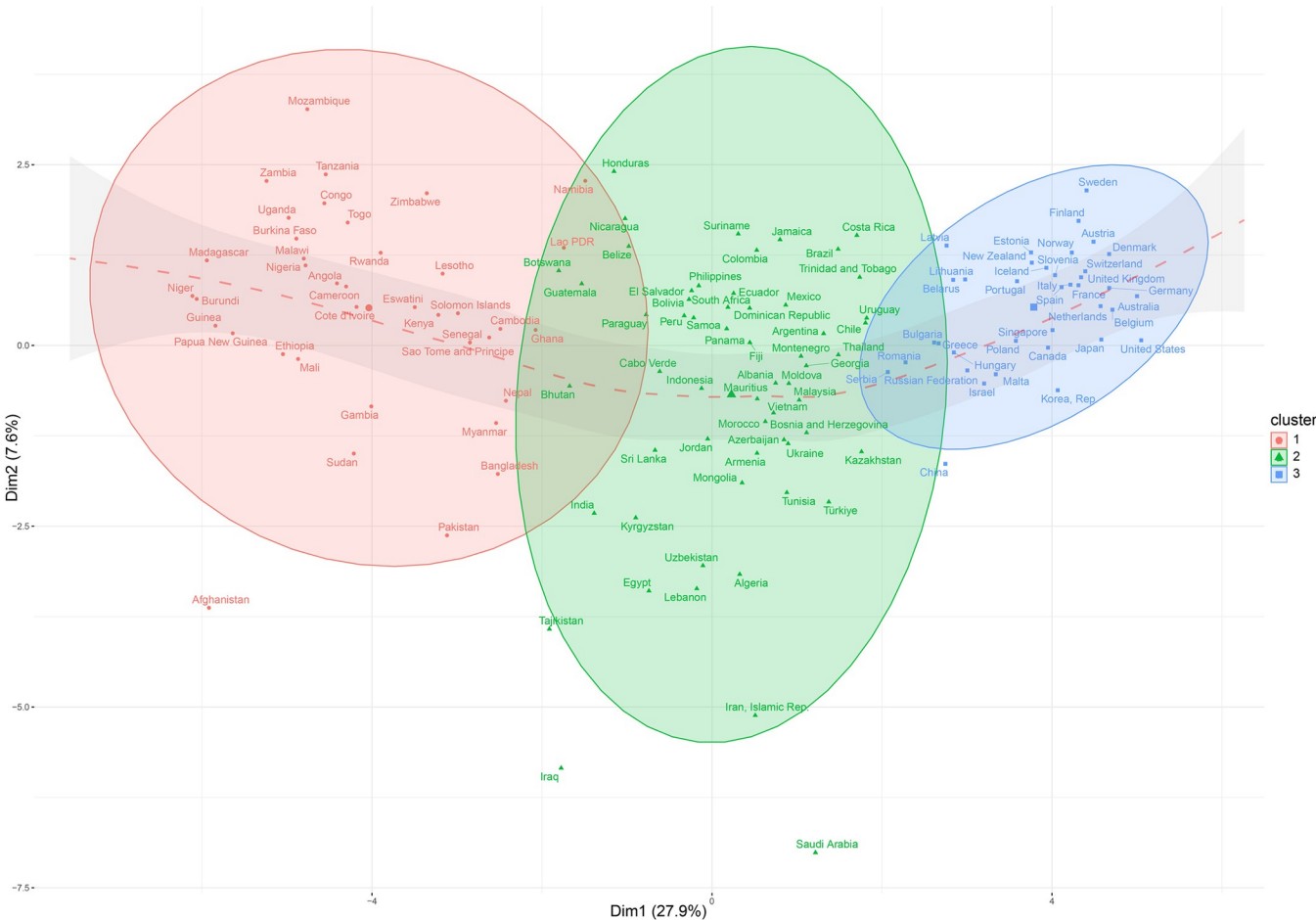

**Fig 4. Projections of countries on the MFA's first plane.** Note: The fitted line was modified to a dashed curve, and the transparency of the confidence interval was increased to enhance the clarity of the figure.

within the Earth's carrying capacity. Moreover, HPI was not significantly correlated with the first PC (0.33, S2 Table), indicating that an increase in GDPP or HDI does not always contribute to ecological well-being.

Fig 4 shows the projections of countries on the MFA's first plane, and the 133 countries were clustered into three groups via the HCPC (S3 Table). In this paper, regional groupings used are based on the "standard country or area codes for statistical use" (M49) (available online: https://unstats.un.org/unsd/methodology/m49/). Besides, World Bank country groups (2023 fiscal year) are divided into low-income (GNI per capita < $1,085), lower middle-income ($1,086 < GNI per capita < $4,255), upper middle-income ($4,256 < GNI per capita < $13,205) and high-income (GNI per capita >13,205) [78]. UNDP country groups (2021 HDI report) are classified by low human development (< 0.550), medium human development (0.550–0.699), high human development (0.700–0.799) and very high human development (≥ 0.800) [79].

Cluster 1 included countries from Sub-Saharan Africa (Burundi, Congo, Ethiopia, Kenya) and Southern Asia (Afghanistan, Bangladesh, Nepal, Pakistan), with an average GDPP and HDI of 1526.04 and 0.54, respectively. According to the World Bank and the UNDP on country classification, most countries in this cluster were classified as low-income to lower-middle-

income and low-human development categories. Cluster 2 included countries primarily from Western Asia and the Gulf (Azerbaijan, Iran, Iraq, Saudi Arabia, Turkey), Central Asia (Kazakhstan, Kyrgyzstan, Tajikistan, Uzbekistan), Southeast Asia (Indonesia, Malaysia, Thailand, Vietnam), Latin America, and the Caribbean (Brazil, Colombia, Costa Rica, Jamaica, Mexico). Their average GDPP and HDI were 6580.72 and 0.75, respectively, suggesting that most countries in this cluster belonged to the upper middle-income and medium to high human development groups. Cluster 3 included countries mainly from North America (Canada and the United States), Australia, New Zealand, and Europe (Poland, Denmark, the United Kingdom, Italy, France, and Germany). Their average GDPP and HDI were 37194.14 and 0.90, respectively, and almost all were developed countries with high income and very high human development. Therefore, from left to right, the GDPP and HDI increased, indicating that the level of economic development is crucial for differentiating country clusters.

Overall, the finding verifies that economic growth could drive most of the other human development dimensions, and this explains why financial markets, economists, media, governments, and companies are persisting with what is known as "GDP fetishism" despite the consensus of a more sustainable "beyond GDP" future [34, 80, 81]. However, the increase in GDPP and HDI are achieved at the expense of high $CO_2$ emissions per capita and higher EFP. This indicates that human progress is based on the mass consumption of natural resources and emissions of greenhouse gases. Developed countries propel their economic growth and welfare based on the overconsumption of non-renewable natural resources and mass greenhouse gas emissions, which are unsustainable [82]. This is because exerting immense pressure on the planet can counteract economic and societal development and adversely influence people's subjective well-being [83, 84].

## 5.2 Economic growth, Kuznets curve, and the environmental Kuznets curve

The second PC explained 7.6% of the total variance, with SDG 10 (reduced inequalities), SDG 15 (life on land), SDG 6 (clean water and sanitation), SDG 5 (gender equality), and SDG 12 (responsible consumption and production) contributing the most (Table 2). SDGs 10 and 5 comprise indicators focusing on general inequality issues, including income and gender inequalities, such as 5.5.2 (proportion of women in managerial positions), 10.4.1 (labor share of GDP), 10.4.2 (Gini index) and 5.5.1 (proportion of seats held by women in national parliaments). SDGs 6, 15, and 12 include indicators regarding climate change, biodiversity loss, and pollution, such as 6.4.2 (level of water stress, freshwater withdrawal as a proportion of available freshwater resources), 12.c.1 (Fossil-fuel subsidies as a proportion of total GDP), 15.1.1 (forest area as a proportion of total land area), and 15.1.2 (average proportion of terrestrial key biodiversity areas (KBAs) covered by protected areas (%)). Moreover, most other SDGs that contribute considerably to the second PC, including SDGs 7, 9, and 11, and their indicators that appear in the top 20 contributors to the second PC, also measure inequality or the environment (Table 3), such as 11.6.2 (annual mean levels of fine particulate matter), 9.4.1 (carbon dioxide emissions per unit of manufacturing value added) and 7.2.1 (renewable energy share in the total final energy consumption). Therefore, the second PC could be defined as inequality and environmental dimensions.

Considering that the first PC is the dimension of economic development, while the second PC represents the dimensions of inequality and the environment, notably, the countries in Fig 4 constitute a curve corresponding to a combined Kuznets curve and environmental Kuznets curve (EKC). Due to the inverse of the vertical coordinate compared to the Kuznets' inverted-U curve, the curve in our case is symmetrical to the Kuznets-type curve but with similar meanings. The Kuznets curve refers to the relationship between income inequality and

economic growth and is characterized by the Kuznets inverted U-shaped curve [85]. Specifically, Kuznets believed that income inequality tends to increase at an early stage of development and then decrease as the economy develops. The environmental Kuznets curve (EKC), introduced by Grossman and Krueger [86, 87], is a hypothesized relationship between various indicators of environmental degradation and per capita income. The EKC hypothesis assumes that environmental degradation increases to a specific level and then decreases with increasing income. As an extension of the Kuznets curve, the EKC has also been characterized as an inverted U-shaped curve.

In our case, the countries in Cluster 1 were classified as having low income and human development levels, mainly from Sub-Saharan Africa and Southern Asia. On average, they performed better in terms of income equality and environmental quality. However, their good performance may be ascribed to their abundant natural resources, widespread poverty, and low levels of industrialization [88]. Cluster 2 comprised the developing countries from Western Asia (especially the Gulf countries), Central Asia, Southeast Asia, Latin America, and the Caribbean with medium income and human development levels but a wide vertical distribution and worse performance in reducing inequality and environmental protection than Cluster 1. Finally, the countries in Cluster 3, primarily from North America and Europe, have the most stable and outstanding performance in achieving high income, controlling inequality, and protecting the environment. Therefore, the nexus between economic growth, income inequality, and the environment raises an interesting but serious question: Can reducing inequality, environmental sustainability, and economic growth coexist in a nation's early development phase? According to the research conducted by Wang and Christensen [17], the experiences of China, Russia, Brazil, and South Africa have provided negative answers.

Since the market-oriented reform in 1978, China achieved a rapid GDP growth of approximately 11.3% by 2020 for over four decades. In 2013, its GDP amounted to 9.57 trillion in the current US$, ranking second after the United States, and in 2021, it reached 17.73 trillion in the current US$ [89]. Many other educational, health, and infrastructure indicators have verified China's social development achievements. For example, between 1990 and 2021, the HDI value improved from 0.484 to 0.768, an increase of 58.7%, ranking at 79 of 191 countries (and territories) and allowing it to enter a high level of human development [90]. However, the miracle of economic growth was achieved at the expense of the deterioration of social and environmental values. Specifically, in the first 30 years after 1978, China transformed from one of the most equal countries in a planned system to one of the most unequal ones in a market system, as measured by the Gini index from 32.2 in 1990 to 43.7 in 2010 [91]. Political corruption has become increasingly severe, ranking 78th in 2010 in the Corruption Perception Index [92]. Moreover, before 2013, environmental pollution was severe, which was mainly reflected in poor air quality in Beijing and the eastern half of China [93, 94]. Therefore, a development path that balances inequality, the environment, and GDP growth, especially in developing countries, is worth exploring and pursuing by governments for a sustainable future.

## 5.3 Governance capacity, a critical enabler of the SDGs

Regarding the third PC, the goals that contributed most were SDG 16 (peace, justice, and strong institutions), SDG 15 (life on land), SDG 5 (gender equality), SDG 6 (clean water and sanitation), and SDG 10 (reduced inequalities) (Table 2). SDG 16 alone contributed to 21.20% of the third PC. Its concept covers various indicators regarding the functioning of institutions and the quality of public policies in tackling issues regarding all three sustainability pillars (economy, society, and environment), such as 16.1.1 (number of victims of intentional homicide per 100,000 population), 6.5.1 (degree of integrated water resources management

implementation), 10.4.2 (Gini index), 5.5.1 (proportion of seats held by women in national parliaments), 15.5.1 (red list index), 16.9.1 (completeness of birth registration), 8.5.2 (unemployment, % of the total labor force), and 8.1.1 (average of the annual growth rate of real GDP per capita between 2015 and 2019) (Table 3). Therefore, this dimension represents governance capacity.

As for the fourth PC, goals that contributed most were SDG 8 (decent work and economic growth), SDG 10 (reduced inequalities), SDG 15 (life on land), SDG 5 (gender equality), SDG 16 (peace, justice, and strong institutions) (Table 2). Notably, SDG 8 contributed 40.79% to this dimension, which was much higher than the other goals. SDG 8 best represents "economic growth" despite that the SDGs attempt to go beyond GDP [10]. Significant indicators in this PC were 8.1.1 (average of the annual growth rate of real GDP per capita between 2015 and 2019), 8.2.1 (average of the annual growth rate of real GDP per employed person between 2015 and 2019), 5.5.1 (proportion of seats held by women in national parliaments), 8.5.2 (unemployment, % of the total labor force), 15.5.1 (red list index), 10.4.2 (gini index), 9.2.1 (manufacturing value added as a proportion of GDP), and 17.1.1 (total government revenue as a proportion of GDP). Indicators 8.1.1 and 8.2.1, representing the economic growth rate, contributed 16.97% and 16.92%, respectively, far exceeding other indicators (Table 3). Hence, the fourth PC could be defined as the economic growth rate.

As demonstrated in the second correlation circle (Fig 3), indicators 8.1.1 and 8.2.1 negatively impacted 15.5.1 (red list index), 16.7.1 (ratio of female members of parliaments), 5.5.1 (proportion of seats held by women in national parliaments). Besides, they showed low correlations with most other indicators, such as 10.4.2 (gini index), 16.1.1 (number of victims of intentional homicide per 100,000 population), 6.5.1 (degree of integrated water resources management implementation), and 8.5.2 (unemployment, % of the total labor force). This again verifies the previous argument that rapid economic growth can hardly coexist with income equality and a pleasant ecological environment, especially for developing countries in the early development phase. For example, southern and southeastern Asian countries that performed best in the fourth PC, such as India, Cambodia, Sri Lanka, and Bangladesh, face severe issues regarding a wide gap in wealth and heavily polluted natural environments [95–97]. Good governance is critical for tackling imbalanced development, primarily by adjusting the priority of public values (PVs), which has been especially stressed in the 2030 agenda.

Governance generally refers to exerting power and functions via a country's economic, social, and political institutions and constitutes a crucial part of the SDGs. Without strong institutions, competent and incorrupt public administrators, and appropriate public policies, it will be impossible to realize the 2030 agenda. Therefore, good governance is an essential enabler of all SDGs [98–100]. As a solid response, the Global Hub on the Governance for the SDGs, a joint OECD-UNDP initiative, was established to facilitate governments in developing fit-for-purpose public governance systems to deliver on the SDGs [101]. To reinforce synergies and minimize trade-offs for sustainable development, forming a holistic view across different PV dimensions and public sectors has become increasingly crucial and stringent, especially regarding the capability to prioritize different PVs in a timely and strategic manner and monitor relevant indicators effectively. Notably, a "Whole of Government" approach that emphasizes the active employ of formal and informal networks across different agencies within the government to increase integration, coordination, and capacity is increasingly critical for SDGs and has been particularly stressed by the UN [102].

China presents a case demonstrating the dynamic prioritization of PVs during its journey from lopsided development to a much more balanced development via integrated governance. To address the imbalanced development resulting from the rapid economic growth, China's political leaders and Communist Party have advocated for the "Five Domains of

Comprehensive Development" since 2012, showing how the PV dimensions of economy, politics, society, culture, and ecology develop synchronously and coordinate with each other [17]. After ten years of holistic and effective governance, China has made evident progress in these five domains, including poverty reduction, environmental protection, and curbing political corruption. For example, the headcount poverty ratio in rural areas has been substantially reduced by 94% from 1980 to 2015, contributing to more than 70% of global poverty reduction over the past four decades [103, 104]. In addition, after a series of anti-corruption campaigns, China's ranking on the Corruption Perception Index increased from 78th in 2010 to 66th in 2021 [105]. According to Wang & Chen [106], the primary logic was to create tremendous economic value in the early phases, enabling China to concentrate on other PV dimensions and achieve sustainable development.

## 5.4 EFP, HPI, and more for sustainable development

As mentioned previously, although EFP and HPI present their novelty in revealing flaws of the SDGs, neither can be a complete metric of sustainability because human development is multi-dimensional that based on plural PVs. Nonetheless, they provide insight that differs from what the SDGs offer.

The EFP measures the need of humans for biocapacity per person, which suggests that the current economy is propelled by the overconsumption of non-renewable natural resources and reveals the unsustainable aspects of the SDGs [82]. Countries performing well in EFP with lower $CO_2$ emissions per capita are mainly those from Sub-Saharan Africa and Southern Asia. However, this does not imply that they outperform developed countries in achieving sustainable development. Instead, they have low levels of income, HDI, and industrialization, which could explain their outstanding performance in the environmental dimension. Therefore, to obtain a sustainable future, we must seek a development path in which GDP grows with lower $CO_2$ emissions and EFP.

The HPI had very low correlations with all SDGs (Figs 2 and 3). Notably, the correlation values of HPI with GDPP and HDI were 0.29 and 0.32, respectively (S1 Fig). Therefore, the HPI provides a distinctive perspective that focuses on environmental pressure-based well-being that the SDGs have not fully captured, namely, increasing well-being through an environmentally sustainable and socially just approach [46, 107]. Therefore, compared to wealthy developed nations with environmentally destructive development patterns, Latin American countries and some Southeast Asia countries lead the HPI ranking, delivering a "sustainable well-being." However, similar to the EFP, the HPI cannot be regarded as a complete sustainability metric. Despite the outstanding performance of ecological well-being values from Latin America and some Southeast Asian countries, they suffer from the deterioration of other PVs, including growth slowdowns, widened income inequality, and severe political corruption. Consequently, many of them have fallen into the "middle-income trap" [108–110]. Realizing the 2030 agenda depends on strong institutions and governance capacity. In our study, many Latin American countries leading the HPI ranking, such as Costa Rica, Ecuador, Jamaica, Brazil, and Mexico, position themselves on the negative axes of the MFA's second plane (S2 Fig), indicating their poor performance in the governance and economic growth rate dimensions.

In summary, EFP and HPI provide us with enlightening information that diagnoses some potential issues and overlooked PVs regarding SDGs, but neither is comprehensive. Development is not static. What matters is to "jump out of the box" and explore new theories and indicators based on PVs, which enables us to enrich our understanding of sustainable development and avoid development failures.

## 6. Conclusion

SDGs are now in a quandary because their theoretical foundations are weak [52, 57]. Consequently, the goals have been criticized for being too ambitious, universal, expansive, and potentially contradictory to planetary limits. Therefore, it is necessary to identify the main dimensions, explore interlinkages, and diagnose potential flaws in the SDGs, as less than half of the time remains to achieve the 2030 agenda. Our study employed MFA and HCPC to investigate these stringent issues and analyzed the results based on the PV theory.

In conclusion, this study yielded four results. First, the HDI and GDPP are closely related to the SDGs and determine country clusters. Despite the "beyond GDP" principle, economic values represented by GDP remain significant in propelling human development, and this explains why "GDP fetishism" is still prevalent nowadays. Our study confirms the contradiction that an increase in income level and HDI is achieved at the expense of high $CO_2$ emissions per capita and higher EFP, with no significant improvements in ecological well-being represented by the HPI. For future sustainable development, SDGs must minimize trade-offs between socioeconomic and environmental goals and proceed within Earth's carrying capacity. Second, the projections of countries in the MFA's first plane approximately correspond to a combined Kuznets curve and an environmental Kuznets curve (EKC). The question of whether economic growth can coexist with environmental sustainability and equality in the early stages of development is a meaningful topic for future research. Third, governance capacity constitutes the third PC and correlates with indicators covering all three pillars of sustainability, serving as a critical enabler of the SDGs, particularly the capability to prioritize different PVs in a timely and strategic manner with national condition changes. Finally, neither EFP nor HPI could reveal the entire story of human development because they overlooked other PVs. However, they provide information that differs from the SDGs. Embracing more such indicators could enrich the theoretical and value bases of development, avoid development failures, and achieve a sustainable future.

Finally, our study has two primary limitations. First, the SDGs remain incompletely measured, and this situation has deteriorated further owing to the COVID-19 pandemic and regional conflicts. Considering that there are only five more years before 2030, it is of great urgency and significance to accelerate the collection of data and ensure the data quality. Insufficiently regularly produced data from many countries (especially sub-Saharan African countries) limited our study to a cross-sectional analysis. Using a subset of indicators may have some potential impacts on the result. However, we believe such impacts would hardly influence our findings and policy suggestions because most indicators from SDGs have medium to high correlations with GDPP or HDI. With more data becoming available in the future, statistical analyses should be repeated to increase the robustness of our research. Another limitation was the selection and definition of PCs. Although 78 PCs were generated from the MFA, we could only select the top four PCs with the most considerable variances because the remaining PCs were not easy to define and interpret, which may be attributed to large data noise derived from data screening and missing data imputation. However, remaining PCs may still have specific meanings that are difficult to define and interpret, which could be a potential orientation for future research.

## Supporting information

**S1 Appendix. Criteria and screening steps used to select indicators and countries.**
(DOCX)

**S1 Table. List of selected SDGs and illustrative groups.**
(DOCX)

**S2 Table. Correlations between SDGs, IGs and first four PCs.**
(DOCX)

**S3 Table. Clusters derived from HCPC.**
(DOCX)

**S1 Fig. Pairwise correlations among IG 1–4.**
(TIF)

**S2 Fig. Projections of countries on the MFA's second plane.**
(TIF)

## Acknowledgments

The authors would like to thank to the anonymous reviewers for helpful comments.

## Author Contributions

**Conceptualization:** Tianchi Chen.

**Data curation:** Tianchi Chen.

**Formal analysis:** Tianchi Chen.

**Funding acquisition:** Bing Wang.

**Methodology:** Tianchi Chen.

**Project administration:** Bing Wang.

**Supervision:** Bing Wang.

**Visualization:** Tianchi Chen.

**Writing – original draft:** Tianchi Chen.

**Writing – review & editing:** Bing Wang.

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
