## [Decision Letter · Decision Letter 0]

24 Jun 2024

PONE-D-24-14863What do the Sustainable Development Goals reveal, and are they sufficient for sustainable development?PLOS ONE

Dear Dr. Chen,

Thank you for submitting your manuscript to PLOS ONE. After careful consideration, we feel that it has merit but does not fully meet PLOS ONE’s publication criteria as it currently stands. Therefore, we invite you to submit a revised version of the manuscript that addresses the points raised during the review process.

We look forward to receiving your revised manuscript.

Kind regards,

Abid Rashid Gill

Academic Editor

PLOS ONE

 [The paper has been supported by the Key Projects of the National Social Science Fundation (23AZD013).].  

4. We note that you have referenced (ie. Husson, F.,  et al. []) which has currently not yet been accepted for publication. Please remove this from your References and amend this to state in the body of your manuscript: (ie “Husson, F., et al. [Unpublished]”) as detailed online in our guide for authors

5. We are unable to open your Supporting Information file [Fig.rar]. Please kindly revise as necessary and re-upload.

6. We notice that your figures are uploaded with the file type 'Supporting Information'. Please amend the file type to 'figure'. Please ensure that each Supporting Information file has a legend listed in the manuscript after the references list.

Reviewers' comments:

Reviewer's Responses to Questions

**Comments to the Author**

1. Is the manuscript technically sound, and do the data support the conclusions?

Reviewer #1: Yes

Reviewer #2: Partly

Reviewer #3: Partly

2. Has the statistical analysis been performed appropriately and rigorously? 

Reviewer #1: Yes

Reviewer #2: No

Reviewer #3: Yes

3. Have the authors made all data underlying the findings in their manuscript fully available?

Reviewer #1: Yes

Reviewer #2: Yes

Reviewer #3: Yes

4. Is the manuscript presented in an intelligible fashion and written in standard English?

Reviewer #1: Yes

Reviewer #2: Yes

Reviewer #3: Yes

5. Review Comments to the Author

Reviewer #1: This paper deals with a very interesting topic.

From my point of view, the paper has important merits, namely:

- the introduction provides, in overall terms, a good motivation for the remaining of the study

- the same applies to literature review, which covers the critical aspects needed as background for the empirical exercise conducted in the empirical section.

-the method is adequate and allows to obtain useful results

Regarding some minor comments, I would like to emphasize the following two aspects:

- perhaps the research questions advanced in the first section could with advantage be relocated to the methodological section

- some improvement in the last section could also be useful, namely providing more critical orientations for future research as well as for practical purposes.

Reviewer #2: 1. The indicators selected in the article are mainly related to economy and development, so it is not surprising to draw relevant conclusions;

2. There have been many empirical studies on Kuznets curve and environmental Kuznets curve in disciplines such as economics, this article only verifies the relevant conclusions;

3.SDGs have been proposed for nearly a decade, so it is certain to have some lag.

In summary, the article lacks significant innovation and is not recommended for adoption.

Reviewer #3: 1. Objectives and Research Questions:

The objectives of the paper are not fully justified and supported by the results. In the discussion section, objectives should connect the findings and discuss their contributions to addressing the research questions.

Clearly state the objectives at the outset and ensure they align with the research questions and methodology. Explain how each objective was intended to contribute to the overall aim of the study.

In the discussion section, explicitly connect each finding to the stated objectives. Discuss how the results contribute to answering the research questions and whether any unexpected findings warrant further investigation or reinterpretation of the objectives.

Identify any gaps or limitations in achieving the objectives based on the findings and propose future research directions to address these gaps.

2. Methodology:

Ambiguity in the selection of SDGs and their indicators, exclusion of SDG 14, and the limited number of indicators used.

Provide a rationale for why specific SDGs and indicators were chosen over others. Consider the relevance to the research questions, data availability, and the ability to provide a comprehensive analysis.

Acknowledge the limitation of using a subset of indicators and justify how these selected indicators are representative of the broader dataset. Discuss the potential implications of excluding certain indicators.

Consider categorizing countries based on multiple criteria, including ecological footprint (EFP) and Happy Planet Index (HPI), in addition to Gross Domestic Product per capita (GDPPC) and HDI. Provide a rationale for the chosen criteria and discuss how these categories align with the research objectives.

3. Results and Discussion:

Confusions between suggested dimensions based on HCPC-selected indicators and the trade-off between socioeconomic and environmental goals. Figures are missing in paper

Clarify how the selected dimensions align with the HCPC-selected indicators and why they are relevant to the research objectives. Discuss any trade-offs identified in the results and how they relate to achieving sustainable development goals.

Address the interlinkages between economic, social, and environmental sustainability beyond CO2 emissions, EFP, GDPPC, and HDI. Consider integrating additional indicators or dimensions to provide a more comprehensive analysis.

Discuss GDPPC as a measure of economic growth and HDI as a measure of human development and well-being. Justify their relevance in assessing sustainable development outcomes and discuss any limitations in their application.

By addressing these points in your paper, you can enhance the clarity, justification, and robustness of your research methodology, results interpretation, and discussion of findings. This approach will ensure that your paper effectively addresses the complexities of sustainability and provides valuable insights into the interplay between economic, social, and environmental factors.

6. PLOS authors have the option to publish the peer review history of their article (what does this mean?). If published, this will include your full peer review and any attached files.

Reviewer #1: No

Reviewer #2: No

Reviewer #3: No

---

## [Author Response · Author response to Decision Letter 0]

6 Aug 2024

Dear Editor:

We have made modifications or explanations to each comment provided by you and reviewers, please see our uploaded file "Response to Reviewers".

Thank you for your consideration. I look forward to hearing from you.

Sincerely,

Tianchi Chen

College of Public Administration, Huazhong University of Science and Technology (HUST)

Wuhan 430074, China

(+86)18021667928

d202081314@hust.edu.cn

---

## [Decision Letter · Decision Letter 1]

26 Aug 2024

What do the Sustainable Development Goals reveal, and are they sufficient for sustainable development?

PONE-D-24-14863R1

Dear  Authors, 

We’re pleased to inform you that your manuscript has been judged scientifically suitable for publication and will be formally accepted for publication once it meets all outstanding technical requirements.

Kind regards,

Abid Rashid Gill

Academic Editor

PLOS ONE

Additional Editor Comments (optional):

Reviewers' comments:

Reviewer's Responses to Questions

**Comments to the Author**

1. If the authors have adequately addressed your comments raised in a previous round of review and you feel that this manuscript is now acceptable for publication, you may indicate that here to bypass the “Comments to the Author” section, enter your conflict of interest statement in the “Confidential to Editor” section, and submit your "Accept" recommendation.

Reviewer #3: All comments have been addressed

2. Is the manuscript technically sound, and do the data support the conclusions?

Reviewer #3: Yes

3. Has the statistical analysis been performed appropriately and rigorously? 

Reviewer #3: Yes

4. Have the authors made all data underlying the findings in their manuscript fully available?

Reviewer #3: Yes

5. Is the manuscript presented in an intelligible fashion and written in standard English?

Reviewer #3: Yes

6. Review Comments to the Author

Reviewer #3: (No Response)

7. PLOS authors have the option to publish the peer review history of their article (what does this mean?). If published, this will include your full peer review and any attached files.

Reviewer #3: No

---

## [Editor Report · Acceptance letter]

4 Sep 2024

PONE-D-24-14863R1 

PLOS ONE

Dear Dr. Chen, 

I'm pleased to inform you that your manuscript has been deemed suitable for publication in PLOS ONE. Congratulations! Your manuscript is now being handed over to our production team.

Kind regards, 

on behalf of

Dr. Abid Rashid Gill 

Academic Editor

PLOS ONE